# Conventional and Conservation Seedbed Preparation Systems for Wheat Planting in Silty-Clay Soil

**Roberto Fanigliulo, Daniele Pochi and Pieranna Servadio \***

Consiglio per la Ricerca in Agricoltura e L'analisi Dell'economia Agraria (CREA), Centro di Ricerca Ingegneria e Trasformazioni Agroalimentari (Research Centre for Engineering and Agro-Food Processing), Via della Pascolare 16, 00015 Monterotondo, Italy; roberto.fanigliulo@crea.gov.it (R.F.); daniele.pochi@crea.gov.it (D.P.)

\* Correspondence: pieranna.servadio@crea.gov.it; Tel.: +39-06-9067-5223

**Abstract:** Conventional seedbed preparation is based on deep ploughing followed by lighter and finer secondary tillage of the superficial layer, normally performed by machines powered by the tractor's Power Take-Off (PTO), which prepares the seedbed in a single pass. Conservation methods are based on a wide range of interventions, such as minimum or no-tillage, by means of machines with passive action working tools which require two or more passes The aim of this study was to assess both the power-energy requirements of conventional (power harrows and rotary tillers with different working width) and conservation implements (disks harrow and combined cultivator) and the soil tillage quality parameters, with reference to the capability of preparing an optimal seedbed for wheat planting. Field tests were carried out on flat, silty-clay soil, using instrumented tractors. The test results showed significant differences among the operative performances of the two typologies of machines powered by the tractor's PTO: the fuel consumption, the power and the energy requirements of the rotary tillers are strongly higher than power harrows. However, the results also showed a decrease of these parameters proceeding from conventional to more conservation tillage implements. The better quality of seedbed was provided by the rotary tillers.

**Keywords:** secondary tillage; machine testing; traction force; fuel consumption; tractor $CO_2$ emissions; tillage quality parameters

## 1. Introduction

The processes of land conversion and agricultural intensification are significant causes of soil quality loss and environmental impact [1]. The influence of soil tillage systems on soil properties and energy efficiency is shown by the important factors of soil fertility conservation and evaluation of agricultural system sustainability [2]. Minimum and zero tillage technologies for soil protection are specific and important ways for resource and energy saving in agriculture [3,4].

Seedbed preparation represents the major cost in crop planting, being one of the most energy-consuming activities [5,6]. The choice of the methods and of the suitable operating machines depends on various factors, such as: the physical characteristics of the soil, the type and intensity of the previous primary tillage, the amount and the type of crop residues present on the surface and especially the type of crop to be sowed. These factors determine the type of seed drill (pneumatic or in-line) to be used and, therefore, the necessity of preparing a seedbed more or less pulverized. Among the many factors that affect the loading of agricultural machinery, the tillage depth is key in the performance evaluation of the tractor because an actual paddy field has different soil properties according to the soil layers. The tillage depth affects the agricultural ecosystem and has a significant impact on crop yields and quality [7].

In Italy, the most common conventional method adopted in silty-clay soils to prepare the seedbed is based on the chopping (rarely the burning) of the residues from previous crops, on a deep ploughing (0.30–0.40 m) aimed at burying the residues [8], and on

secondary tillage by means of machines with working tools operated by the tractor's Power Take-Off (PTO), such as power harrow or rotary tiller [9,10]. Unwanted effects of such methods can be excessive energy requirements [11], and related costs [12], due to high PTO torque needed to drive the rotors, worsening of soil structure due to compaction which leaves the soil susceptible to crusting and affects plant growth and yield, loss of nutrients in deeper layers, mineralization of organic matter in upper layers, increasing soil erosion caused by wind and runoff [13]. Moreover, much research has shown that high working speed and deep tillage results in greater tractor's $CO_2$ emissions [14,15], due to high fuel consumption [16], and favourable conditions for microbiological activity created by the deep tillage [17]. Koga et al. [18] established that 45% of $CO_2$ emissions were from tractors operating in wheat production. Reducing fuel consumption in seedbed preparation has become increasingly important since other studies have established that the emission of $CO_2$ from the consumption of tractor fuel is equivalent to 276 kg $CO_2$ per 100 l diesel [19,20].

Conservation tillage methods are based on a wide range of interventions [21–24], that contribute to preserving soil fertility through the reduction of the number of machine passes and of working depth, such as the soil tillage without inversion of the layers [25] and the contemporary superficial secondary tillage, by means of combined implements [26], and the minimum tillage, with cultivators and disk harrows [27]. These machines perform soil fragmentation by absorbing only the tractor drawbar power. Furthermore, the need to also reduce both the operative working time and the soil compaction, due to the repeated passages of the machines, directed the attention of the agricultural operators towards one pass combined machines characterized by wide working width (5–7 m) and tillage tools with suitable geometry for different soil conditions [28].

Many field studies have been carried out to compare these tillage methods for seedbed preparation. Ghuman and Sur [29] studied the effects of three different systems on wheat for five years: minimum tillage with residue, minimum tillage without residue and conventional tillage. The grain yield in the minimum tillage with residue treatment remained below the conventional tillage treatment during the first two years but it was greater in the next three years. Papayiannopoulou et al. [30], compared conventional cultivation and conservation tillage of winter cereals (wheat) in clay soil. Due to the reduction in the number of the cultivations interventions, obtained results showed 40% of fuel-saving, which, combined with reduced labour and machinery needs, contributes to significant economic benefits.

To evaluate the seedbed quality, soil coverage by crop residues, surface roughness and cloddiness in the tilled layer were measured [31]. Generally, conventional primary tillage increases soil roughness and cloddiness when compared to the no-tillage system or minimum tillage [32,33]. A uniform soil surface, well-pulverized seedbed and evenly distributed residue are key to achieving perfect planting conditions [34,35]. Soil roughness is a measure of the variations in surface elevation, as the aggregate of peaks and depressions present on the soil surface, which may range from few millimeters to several decimeters. Regarding cloddiness, the action of the working tools of soil tillage machines determines the breaking of the soil with the formation of clods, of variable dimensions depending on the tillage system and working speed [36]. Excessive soil surface roughness and cloddiness in the tilled layer requires further interventions by other operating machines, which increase production costs and degrade the soil structure (crusting).

The objectives of this study were to choose the best energy-efficient implement, within conventional and conservation cereal production systems, to evaluate the soil tillage quality parameters, especially with reference to the capability of preparing an optimal seedbed for wheat planting, as a function of tractor speed, tillage depth and harrows and tillers rotors rotational speed [37–39].

## 2. Materials and Methods

The Agricultural Machinery Testing Centre at CREA performed tests to measure and compare the energy requirements of seven conventional implements: four power harrows, with working tools rotating on a vertical axe, three rotary tillers, with tools rotating on a horizontal axe, and two conservation implements (offset disk harrow and combined cultivator) with passive action working tools. These implements are commonly used for the secondary tillage of the silty-clay soil, widespread in Central Italy. The tests were carried out in accordance with a specific protocol proposed by ENAMA (Italian Agency for Agricultural Mechanization), drawn up by national experts in the sector of agricultural engineering and based on the current international reference standards (EN, ISO, ASABE). This protocol, characterized by methodological strictness, defines, in detail, the methodology to be followed in performing field tests. Moreover, the test protocol was recognized in Europe through the ENTAM (European Network for Testing of Agricultural Machinery) which represents a network of as many as 11 European test centers that have signed an agreement for the harmonization of test procedures and the mutual recognition of test reports.

### 2.1. Tested Implements and Tractor Characteristics

The selected implements are commonly used after deep ploughing (in secondary tillage), aimed at cereals planting (Figure 1). In the following, the power harrows and rotary tillers are identified both by the initials (PH and RT respectively), and by the working width (m) for a total of 9 treatments.

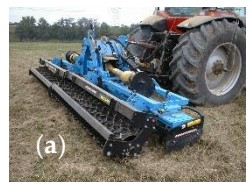 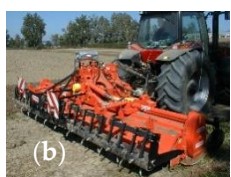 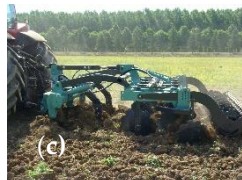 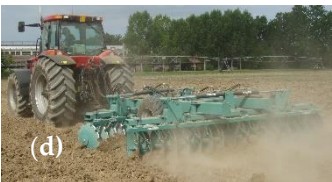

**Figure 1.** Selected implements during the tests. (**a**) 5 m power harrow; (**b**) 5.2 m rotary tiller; (**c**) combined cultivator; (**d**) offset disk harrow.

The main technical data of the tested machines are reported in Tables 1 and 2.

**Table 1.** Main technical data of the tested power harrows (PH).

| Specifications/Treatments | PH-3 | PH-4 | PH-5 | PH-6 |
|---|---|---|---|---|
| Working width (m) | 3.0 | 4.0 | 5.0 | 6.0 |
| Numbers of rotors | 12 | 16 | 20 | 24 |
| Distance among rotors (mm) | 245 | 245 | 245 | 245 |
| Rotor tine length (mm) | 280 | 270 | 260 | 310 |
| Packer roller diameter (mm) | 500 | 500 | 470 | 500 |
| Total mass (kg) | 1320 | 1650 | 2910 | 4000 |

The power harrows (PH) act the soil secondary tillage by means of a number of tines mutually counter-rotating on the vertical axis. The machines were endowed with a rear levelling bar, to uniform the tilled soil by the tines. The rotary tillers (RT) were equipped with L-shaped blades flanged on a horizontal axis. A rear carter performed the fragmentation of the soil slice cut by the blades. Both machine typologies were equipped with a rear roller to adjust the working depth and to level the surface of the tilled soil. The combined cultivator (CC) consisted of a shanks subsoiler followed by two series of disks with an opposite inclination for soil reversing and mixing with crop residues; an innovative rear roller operated the soil surface leveling, further refining the soil surface and the control of the working depth. The offset disk harrow (DH) consisted of two gangs of concave disks angled in opposite directions. The disks of the front gang were notched

and those of the rear gang were plain. The disk's diameter was 0.66 m. The gangs were set at the maximum angle (22.5°) to guarantee the maximum tillage aggression. A hydraulic actuator provided the control of the angle between each gang, while a hydraulic height adjustment of the frame transport wheels controlled the set tillage depth.

**Table 2.** Main technical data of the tested rotary tillers (RT) and of the conservation implements (disks harrow, DH, and combined cultivator, CC).

| Specifications/Treatments | RT-2.3 | RT-4 | RT-5.2 | DH | CC |
|---|---|---|---|---|---|
| Working width (m) | 2.3 | 4.0 | 5.2 | 2.5 | 3.9 |
| Blades/tools number | 54 | 96 | 120 | 36 | 5 [1]–10 [2] |
| Distance among flanges/tools (mm) | 250 | 240 | 250 | 230 | 950 [3]–480 [4] |
| Number of blades on a flange (n) | 6 | 6 | 6 | - | - |
| Rotor diameter [5] (mm) | 530 | 600 | 550 | - | - |
| Total mass (kg) | 1170 | 2560 | 3140 | 3465 | 1730 |

[1] Shanks number; [2] Disks number; [3] Distance among shanks; [4] Distance among disks; [5] working tools included.

Two different engine power tractors were used during the tests: (1) a 4 WD tractor (Landini Legend 145, Fabbrico (RE), Italy), with engine nominal power of 110 kW and total mass of 6420 kg, for testing the 3.0 m power harrow and the 2.3 m rotary tiller. (2) a 4 WD tractor (Case IH MX 270, Racine, WI, USA), with the power of 205 kW and a total mass of 11,000 kg, to test all other machines. To reduce the tractor's wheel slippage, a front ballast (420 and 630 kg respectively) was hooked. The tractors were turned on at the minimum regime one hour before the in-field tests and engaging its rear PTO to warm up the tractor fluids and the implements lubricants. All tests were performed with diesel fuel in compliance with the EN 590 [40], and it was always provided by the same supplier. Consequently, its quality was assumed to be constant, with a Low Heating Value of 42.7 MJ kg$^{-1}$. $CO_2$ emission calculation was based on the tractor's fuel consumption and the datum that the combustion of 1.0 l diesel oil results in the emission of 2.76 kg $CO_2$.

## 2.2. Field Site

The tests were carried out, over more years, at the experimental farm of CREA in Monterotondo (Rome, Central Italy; 42°5′51.26″ N; 12°37′3.52″ E; 24 m a.s.l.), on flat plots (<1% slope), silty-clay soil (clay 54.3%, silt 43.4%, sand 2.3%) according to the USDA soil classification system. The 9 test plots were ploughed at about 0.30 m depth to incorporate plant residues. Before the ploughing, the following characteristics and parameters were measured (ten measurements, on a diagonal within each test plot) in the soil layer corresponding to the working depth: water content and dry bulk density, soil coverage index (SCI) by crop residues and/or biomass, resistance to penetration (Cone Index). The first two parameters were calculated from soil samples of 100 cm$^3$ extracted at different depths by means of a manual soil coring tube (Eijkelkamp, Giesbeek, The Netherlands), and dried in an oven at 105 °C until the mass becomes constant [41,42]. The SCI was determined by means of image analysis. A 1 m$^2$ square frame was placed on the ground and a digital image of the soil confined by the frame was taken by means of a Nikon camera and stored. Then, through the software Adobe Photoshop (Adobe Systems Software, Dublin, Ireland), the percentage of soil area covered by residue was detected and quantified. The Cone Index was determined according to the ASAE standard S313.3 [43], by means of a hand-operated penetrologger (Eijkelkamp, Giesbeek, The Netherlands), measuring the resistance to penetration of the undisturbed soil. The instrument provided a detailed vertical profile of soil strength.

## 2.3. Operating Machinery and Soil Parameters

According to the ENAMA (Italian Agency for Agricultural Mechanization) test protocol [44], the following dynamic–energetic parameters of each tractor–machine system were measured: width and depth of tillage, speed, time and capacity of work; PTO torque,

speed and resulting power; force of traction and resulting power; tractor's power, slip and corresponding losses; fuel consumption and energy required per surface unit and per volume unit of tilled soil.

Before the tests, the tractor's engine performances were verified at the dynamometric brake (Borghi and Saveri, Bologna, Italy) that provided the updated characteristic curves. The dynamometric brake was also used, after the field tests, in order to reproduce the mean conditions of fuel delivery, measured PTO and engine speed. This simulation aimed at evaluating the total torque and power provided by the engine and the corresponding fuel consumption [45].

Multiplying the total engine power ($W_t$, kW) by the actual working time ($T_e$, h ha$^{-1}$), will provide the energy required per surface unit area (Equation (1)):

$$E_{ha} \text{ (MJ ha}^{-1}) = 3.6 \cdot W_t \cdot T_e \tag{1}$$

Dividing $E_{ha}$ by the working depth (P, m), will give the energy per unit of volume of tilled soil ($E_{vol}$) (Equation (2)):

$$E_{vol} \left( \text{MJ } 10^{-3}\text{m}^{-3} \right) = \frac{E_{ha}}{10 \, P} \tag{2}$$

The balance of the power delivered by a tractor is described by the relation (3):

$$W_{eng} \text{ (kW)} = W_{trs} + W_{sd} + W_{tr} + W_{pto} + W_s \tag{3}$$

where: $W_{eng}$ = power at engine drive shaft; $W_{trs}$ = power dissipated by the transmission of motion to wheels and PTO; $W_{sd}$ = power required by tractor self-dislocation; $W_{tr}$ = traction power (kW); $W_{pto}$ = PTO power (kW); $W_s$ = power lost for slip.

Considering the second member of relation (3), in a field test, all components except $W_{trs}$ are measured directly net of transmission losses. These occur between the crankshaft and the wheels and/or PTO and represent, in fact, the $F_{trs}$ component and can be calculated by knowing the transmission efficiency, $\eta$, that in our case is assumed $\eta$ = 0.87. Therefore, relation (3) can be modified as follows:

$$W_{eng} \text{ (kW)} = (W_{sd} + W_{tr} + W_{pto} + W_s))/\eta \tag{4}$$

Similarly, all data provided by the test at the dynamometric brake are based on PTO measurements of torque and power and the crankshaft values can be assessed by dividing them by $\eta$.

As to the component $W_s$, a good assessment of its average is provided the relation (5):

$$W_s(\text{kW}) = F_{tr} \times \Delta v \tag{5}$$

where: $F_{tr}$= average force of traction (daN) measured during the work; $\Delta v$ = difference between the average tractor wheels peripheral speed (m s$^{-1}$) during the work and the average actual working speed (used to calculate the average slip).

As to the quality of tillage, it was evaluated through the determination of the following parameters: crop residues/biomass burying degree (BBD), soil surface roughness index (SRI), roughness reduction degree (RRD), clod-breaking index (CBI), cloddiness reduction degree (CRD) and seedbed quality index (SQI) [46]. They were measured in ten random points in each test. The BBD is calculated from the values of the SCI determined before and after the implement tillage by means of Equation (6).

$$BBD \, (\%) = 100\frac{(SCI_{us} - SCI_{ts})}{SCI_{us}} \tag{6}$$

where $SCI_{us}$ is the soil coverage index of undisturbed soil and $SCI_{ts}$ is the index of tilled soil.

The SRI and the working depth were determined, transversally to the direction of work, by means of an in-house designed profile meter. This instrument consists of a laser sensor (Leica Geosystem Disto, Heerbrugg, Switzerland) which measures its distance from the ground as it moves on a horizontal aluminum rail placed above the tilled strip, by means of a step-by-step electric motor, drawing the surface profile of the ground [47]. The rail is fixed on two tripods and its length must suffice to cover the entire working width of the machines to be tested. A personal computer, placed on a manually moved trolley, collected and processed the data (interval between two readings: 10 mm), by means of a software program (in Microsoft Visual Basic 6.0) which controls the movement of the laser probe and the sampling rate per unit of distance. The profile meter was powered by a 12 V battery. The relief of the profiles, in the same point, before and after the execution of an operation (e.g., secondary tillage), provided, respectively, the roughness indices $\sigma_{r1}$ and $\sigma_{r2}$ (based on the calculation of the standard deviation of the detected heights series). These indices enable the calculation of the roughness reduction degree, RRD, resulting from the secondary tillage, as follows (Equation (7)):

$$\text{RRD} \ (\%) = 100 \frac{(\sigma_{r1} - \sigma_{r2})}{\sigma_{r1}} \tag{7}$$

The measurement of the cloddiness was made by digging a 0.5 m square trench to the working depth. The soil aggregates found in this volume, were left to dry for at least 20 min and weighed on a balance, then they were divided into six size classes by means of hand-operated standard sieves. An index ($I_{ai}$), ranging from 0 for the biggest class to 1 for the smallest class, is attributed to each class [48]. The cloddiness was evaluated by observing the percent of each size class mass referred to the sample total mass. From the cloddiness, the CBI ($I_a$) is calculated as follows (Equation (8)):

$$\sum_{i=1}^{6} \frac{M_i \cdot I_{ai}}{M_t} \tag{8}$$

where $M_i \cdot I_{ai}$ is the product of the index assigned to a clod size class and the mass (kg) of soil belonging to the same class; $M_t$ is the total mass of the sample (kg).

Comparing the CBI values observed before ($I_{a1}$) and after ($I_{a2}$) the secondary tillage, will provide the CRD by means of Equation (9).

$$\text{CRD} \ (\%) = 100 \frac{(I_{a2} - I_{a1})}{I_{a2}} \tag{9}$$

The quality of the seedbed is assessed based on the cloddiness values observed after the passage of the implements. It is described by the SQI, by means of Equation (10):

$$\text{SQI} = \frac{M_{\varnothing \leq 10}}{M_{\varnothing > 10}} \tag{10}$$

where $M_{\varnothing \leq 10}$ is the mass (kg) of the clods with a diameter less or equal to 10 mm and $M_{\varnothing > 10}$ is the mass (kg) of the clods with a diameter over 10 mm.

To evaluate the tillage intensity of the rotary tillers, due to the action on the soil of the L-shaped blades, the coefficient of velocity ($\lambda$), the tillage pitch (P) and the cutting density (CD) were also calculated [49]. The first represents the ratio between the blades-holder rotor angular velocity and the machine working speed and was calculated by the Equation (11):

$$\lambda = \frac{\omega \, r}{v} \tag{11}$$

where: $\omega$ = rotor angular velocity (rad s$^{-1}$); r = rotor radius (m); v = working speed (m s$^{-1}$).

The tillage pitch expresses the distance between two successive cuts of the blades on the same side of the flange, where they are bolted, and was calculated using the following formula (12):

$$P\ (mm) = \frac{4\ \pi\ r}{\lambda\ z} \tag{12}$$

where z is the number of blades on each flange.

The cutting density takes into account the tillage pitch and the distance between contiguous flanges on the rotor and expresses the number of the cuts in a 1 m$^2$ of tilled soil (Equation (13)):

$$CD\ \left(n\ m^{-2}\right) = \frac{2}{P\ d} \tag{13}$$

where d is the distance between the rotor flanges.

Regarding the power harrows tillage intensity, we calculated the tillage pitch of the cycloidal trajectory described by the rotary tines during their action in the soil, with respect to the horizontal plane, by means of the following Equation (14):

$$P\ (mm) = \frac{v}{\omega}\ 2\pi\ 10^3 \tag{14}$$

where: v = working speed (m s$^{-1}$); $\omega$ = rotor angular velocity (rad s$^{-1}$).

The thus defined tillage pitch, divided by the number of the tines of each vertical rotor, provides a dimension called "tines cutting interval" (mm), which could well indicate the tillage strength [50,51].

### 2.4. Measurement Equipment and Data Acquisition System

The data of the most significant operating parameters of each tractor–implement system were collected by an integrated system based on two units, a field unit, and a support unit [52,53], fully assembled at CREA. The tractor (equipped with transducers and a personal computer with a PCI card for real-time data acquisition and an LCD monitor) and a photocells system (placed in each test plot and indicating the start and stop of the test) represent the field unit. The transducers' signals were recorded at a scan rate of 10 Hz. The support unit consists of a van equipped as a mobile laboratory. Its PC is in communication with the field unit's PC by means of a radio–modem system, exchanging data and allowing to monitor the behaviour of critical parameters and the efficiency of the transducers. The support unit also lodges the equipment and instruments used in the evaluation of the quality of work.

As reported by Pochi and Fanigliulo [54], in the tests with passive action implements, the following instrumental system was used: a digital encoder (Tekel TK510, Turin, Italy), mounted on a rear wheel of the tractor measured wheel revolutions on a reference distance, allowing calculation of travel speed under tractor self-displacement in work conditions, and slip, and two mono-axial load cells (AEP Transducers TC4, Modena, Italy) with full scales respectively of 98 kN (tests with disk harrow and combined cultivator) and 49 kN (tests with power harrows and rotary tillers), measuring the force of traction as follows. In traction tests, the load cell is lodged in a drawbar, properly designed to protect it from transversal stresses. Such a drawbar connects a traction vehicle to the tractor–implement system. This is pulled, with gear in "neutral", with the working speed set in the actual tillage with the same implement simultaneously: this test executed with implement working will provide the gross traction force. Repeating the test with implement raised will provide the force required by the self-displacement of the tractor–implement system. The net fraction force will result in the difference between said values. The traction vehicle can be considered as a further element of the field unit. Its PC transmits the traction data to the support unit.

For tillage with rotary implements, in addition to the sensors first mentioned, the tractor's PTO was equipped with a torque meter (HBM T10F/FS, Darmstadt, Germany), with a full scale of 3 kNm, for the measurement of the torque and speed during the work, necessary for the calculation of the required power.

Preliminary tests were conducted with the aim of finding the most correct adjustment of each tractor–implement system. The tractor worked with locked differential and under maximum fuel delivery conditions. Working speeds and depths were set considering soil physical–mechanic characteristics and tillability (according to water content). Each test was replicated three times. The experiment was carried out following a randomized distribution of the plots treated with each secondary tillage method. The plots were 100 m long and 20 m wide. The power harrows, the rotary tillers and the disk harrow experiments followed mouldboard ploughing, while the combined cultivator experiment was conducted in undisturbed soil. In harrowing tests, the tillage depth was set to 0.15 m, while in the rotary tilling to 0.20 m. The conservation implements worked at medium tillage depth, depending on soil conditions.

### 2.5. Statistics

The statistical elaboration aimed at verifying the presence of any general trend statistically significant in the results of tests characterized by great heterogeneity of machine types and dimensions interacting with the variability of soil conditions. The Principal Component Analysis (PCA) was firstly used to observe the relationship among the parameters describing the soil conditions and the performance of the machines. In order to reduce the variability due to the machine's dimensions, the measured dynamic–energetic parameters (power, energy) were used to calculate the relative "specific parameters" (which refer to the unit of surface, of working width, of tilled section surface, etc.). The parameters considered in the PCA are reported in Table 3.

**Table 3.** Parameters used in the PCA.

| | Variables | Unit |
|---|---|---|
| A | Average Cone Index (0–0.30 m) | MPa |
| B | Dry bulk density | $kg\,m^{-3}$ |
| C | Water content, weight percentage | % |
| D | Fuel consumption per h per working width unit | $kg\,h^{-1}m^{-1}$ |
| E | Specific traction force | $kN\,dm^{-3}$ |
| F | Specific traction power | $kW\,dm^{-3}$ |
| G | Tractor slip | % |
| H | Slip Power losses | kW |
| I | Energy losses for slip | $MJ\,ha^{-1}$ |
| J | Total working specific power | $kW\,m^{-1}$ |
| K | Working energy per surface unit | $MJ\,ha^{-1}$ |
| L | Energy losses for transmission | $MJ\,ha^{-1}$ |
| M | Total energy losses | $MJ\,ha^{-1}$ |

The PCA was carried out separately for the four power harrows (PH) and the three rotary tillers (TH) on the normalized data. Based on the indications provided by the PCA, the correlation coefficients between the soil parameters (A, B, C) and the performance-specific parameters (D to M) were calculated together with the related probability of uncorrelation. Eventually, an ANOVA was made to compare the PH and RT datasets.

The test regarded some of the specific parameters described above. This allowed us to consider the data of each machine as a replicate.

The PH-6 data were excluded from the analysis in order to have in both datasets three replicates with working widths as much as possible homogeneous. The statistical elaborations were made by means of the software Past 4.05 [55]. Due to insufficient availability of data, the results of the tests on the combined cultivator (CC) and the disk harrow (DH) were not considered in the statistical elaboration.

## 3. Results and Discussion

### 3.1. Dynamic–Energetic Performance

The tests were aimed at evaluating the aspects of both in-field performance and quality of work of four power harrows and three rotary tillers, with different working width, and of two implements with passive action working tools.

Table 4 shows the results of the measurements of the parameters which describe the dynamic–energetic performances of the nine tractor–machine coupling. The table also reports a series of "specific parameters" (Table 3), used in the statistical elaboration of the results, and the average values of the physical–mechanic characteristics of the soil.

**Table 4.** Average values of the physical–mechanic characteristics of the soil and of the parameters describing the technical performances of the tested machines.

| Parameters | Unit | PH-3 | PH-4 | PH-5 | PH-6 | RT-2.3 | RT-4 | RT-5.2 | CC | DH |
|---|---|---|---|---|---|---|---|---|---|---|
| Water content, weight % | % | 11.0 | 12.8 | 14.3 | 11.9 | 17.2 | 10.7 | 15.1 | 11.6 | 16.7 |
| Dry bulk density | $kg\,m^{-3}$ | 1309 | 1390 | 1420 | 1380 | 1200 | 1100 | 1376 | 1530 | 1599 |
| Surface coverage index | % | 3.9 | 3.6 | 3.4 | 2.9 | 2.9 | 3.6 | 3.9 | 7.6 | 86.1 |
| Average Cone Index (0–30 cm) | MPa | 1.3 | 1.3 | 0.6 | 1.3 | 1.7 | 1.1 | 0.8 | 2.0 | 2.0 |
| Actual working width | m | 3.1 | 4.0 | 5.0 | 6.0 | 2.3 | 4.0 | 5.2 | 2.5 | 4.0 |
| Working depth | m | 0.2 | 0.2 | 0.2 | 0.2 | 0.2 | 0.2 | 0.2 | 0.3 | 0.2 |
| Surface of tilled soil section | $dm^2$ | 45.8 | 64.3 | 75.0 | 96.0 | 46.0 | 88.0 | 109.2 | 83.1 | 59.7 |
| Actual working speed | $km\,h^{-1}$ | 3.5 | 3.6 | 3.4 | 2.7 | 2.9 | 4.0 | 3.3 | 4.7 | 6.3 |
| Tractor wheels peripheral speed | $km\,h^{-1}$ | 3.5 | 3.7 | 3.5 | 2.7 | 3.0 | 4.1 | 3.4 | 5.29 | 6.9 |
| $\Delta v$ | $km\,h^{-1}$ | 0.1 | 0.3 | 0.1 | 0.1 | 0.1 | 0.1 | 0.1 | 0.6 | 0.5 |
| Tractor slip | % | 2.3 | 7.3 | 3.6 | 7.8 | 0.9 | 1.2 | 1.6 | 12.1 | 7.7 |
| Actual working time | $h\,ha^{-1}$ | 0.9 | 0.7 | 0.6 | 0.6 | 1.5 | 0.6 | 0.6 | 0.9 | 0.4 |
| Actual working capacity | $ha\,h^{-1}$ | 1.1 | 1.5 | 1.7 | 1.6 | 0.7 | 1.3 | 1.5 | 1.1 | 2.5 |
| Fuel consumption per h | $kg\,h^{-1}$ | 18.5 | 22.8 | 34.8 | 36.1 | 26.8 | 35.8 | 44.6 | 24.3 | 19.8 |
| Fuel consumption per ha | $kg\,ha^{-1}$ | 17.6 | 15.5 | 20.5 | 22.7 | 40.2 | 27.5 | 29.7 | 21.3 | 7.8 |
| Gross traction force | kN | 13.1 | 18.9 | 24.7 | 22.9 | 8.9 | 14.0 | 20.3 | 49.0 | 24.9 |
| Tractor self-dislocat. force | kN | 7.1 | 7.7 | 12.9 | 9.8 | 7.8 | 11.0 | 12.9 | 6.4 | 5.9 |
| Net traction force | kN | 6.0 | 11.2 | 11.9 | 13.2 | 1.1 | 3.0 | 7.4 | 42.6 | 19.0 |
| Specific traction force | $kN\,dm^{-2}$ | 0.1 | 0.2 | 0.2 | 0.1 | 0.0 | 0.0 | 0.1 | 0.5 | 0.3 |
| Tractor self-dislocat. power | kW | 6.8 | 7.8 | 12.1 | 7.2 | 6.3 | 12.2 | 11.8 | 8.3 | 6.4 |
| Traction power | kW | 5.7 | 11.1 | 11.1 | 9.5 | 0.9 | 3.3 | 6.8 | 55.0 | 33.4 |
| Specific traction power | $kW\,dm^{-3}$ | 1.9 | 2.8 | 2.2 | 1.6 | 0.4 | 0.8 | 1.3 | 22.4 | 8.4 |
| PTO speed | $rad\,s^{-1}$ | 102.3 | 106.8 | 112.0 | 118.5 | 108.9 | 111.0 | 117.3 | - | - |
| Torque at the PTO | Nm | 356 | 385 | 860 | 992 | 726 | 1078 | 1356 | - | - |
| Power at the PTO | kW | 36.4 | 41.1 | 96.3 | 117.6 | 79.0 | 126.6 | 159.0 | - | - |
| Specific PTO power | $kW\,dm^{-3}$ | 11.9 | 10.2 | 19.3 | 19.6 | 34.3 | 31.7 | 30.6 | - | - |
| Total working power | kW | 42.1 | 52.2 | 107.4 | 127.1 | 79.9 | 129.9 | 165.8 | 61.7 | 33.4 |
| Total working specific power | $kW\,m^{-1}$ | 13.8 | 13.0 | 21.5 | 21.2 | 34.7 | 32.5 | 31.9 | 25.2 | 8.4 |
| Total engine power | kW | 56.2 | 70.2 | 137.8 | 144.0 | 95.8 | 145.3 | 191.2 | 84.5 | 53.4 |
| Power losses for slip | kW | 0.1 | 2.9 | 1.0 | 0.8 | 0.02 | 0.3 | 0.6 | 6.64 | 2.56 |
| Energy losses for slip | $MJ\,ha^{-1}$ | 0.5 | 7.2 | 2.0 | 1.9 | 0.11 | 0.63 | 1.21 | 21.0 | 3.7 |
| Working energy per surface unit | $MJ\,ha^{-1}$ | 143.9 | 128.5 | 227.5 | 287.7 | 431.2 | 292.2 | 347.8 | 267.0 | 76.2 |
| Working energy per 1000 $m^3$ soil volume | $MJ\,10^{-3}m^{-3}$ | 96.0 | 80.3 | 151.7 | 179.8 | 215.6 | 132.8 | 165.6 | 78.8 | 50.8 |
| Power losses for transmiss. | kW | 4.8 | 9.1 | 17.9 | 10.1 | 9.6 | 10.2 | 13.4 | 5.9 | 5.6 |
| Energy losses for transmiss. | $MJ\,ha^{-1}$ | 16.5 | 22.4 | 37.9 | 22.8 | 51.7 | 22.9 | 28.1 | 18.7 | 8.0 |
| Total energy losses | $MJ\,ha^{-1}$ | 16.6 | 25.3 | 38.9 | 23.6 | 51.7 | 23.2 | 28.6 | 25.35 | 10.7 |
| Rotor speed | $rad\,s^{-1}$ | 29.6 | 36.5 | 37.6 | 39.8 | 19.8 | 27.7 | 28.6 | - | - |
| Tines cutting interval | mm | 101.7 | 87.0 | 77.8 | 58.0 | - | - | - | - | - |
| Tillage pitch | mm | - | - | - | - | 85.0 | 83.8 | 67.0 | - | - |
| Blades cutting density | $n\,m^{-2}$ | - | - | - | - | 94.1 | 99.4 | 119.3 | - | - |



The average values of the measured tillage depths were close to the set values in all tested machines. As regards the performances of the four power harrows (PH), at increasing working width and total power provided by the tractor's engine, the fuel consumption, the force of traction, and the energy required per surface unit and per volume unit increased as well, while the tines cutting interval decreases.

The average PTO torque, required by the PH rotors and affected by working speed and depth, varied from 356 Nm of the PH-3 to 992 Nm of the PH-6 resulting in increasing power at the tractor's PTO.

Considering the working speed adopted values, limited differences were observed in the values of working time and capacity, which, however, increase with the working width. Moreover, the tractor's slip, and the corresponding energy losses remained at acceptable levels for the type of tillage.

As to the rotary tillers (RT), the fuel consumption per hour and the force of traction increased with the working width. The RT-2.3 showed higher energy requirements per surface unit and per volume of tilled soil, with respect to the two other tillers, due to the lower working speed and to the higher working time. The traction force required by these machines ranged from 1.1 kN for the RT-2.3 to 7.4 kN for the RT-5.2 m. Compared to the power harrows, the rotary tillers required lower traction force, with consequent negligible slip values, due to the forward push action exerted by the L-shaped working tools on the ground, facilitated by its toughness. The average torque at the PTO, required by the rotary tillers drive shaft, varied from 726 Nm of the RT-2.3 to 1356 Nm of the RT-5.2, with corresponding power at the PTO ranging from 79.1 kW to 159.0 kW. The torque and power high requests and variability were probably affected by the variability of the soil conditions, mainly the dry bulk density and the cone index.

The test results showed a higher average power per surface unit (MJ ha$^{-1}$), required by the three rotary tillers compared to the four power harrows, about 36%. The power harrows, to assure the same reduction effect, need a rotation speed about 23% superior to one of the rotary tillers.

Regarding the relationship between tillage pitch and required PTO power, both for power harrows and rotary tillers, as the power increased, the "tines cutting interval" and the "tillage pitch" decreased, from 101 to 58 mm and 85 to 67 mm respectively, while the "blades cutting density" of the rotary tillers increased (from 94 to 119 N m$^{-2}$).

Comparing the results of the 4.0 m width rotary tiller with the power harrow with the same working width (Table 4), the harrowing seems to represent a valid alternative to rotary tilling, allowing a considerable reduction of the energy requirements per soil surface unit (34%), per volume unit of tilled soil (9%), and of the fuel consumption per hour (32%).

Considering the four power harrows fuel consumption (Table 4), the corresponding $CO_2$ emissions were 39.1, 47.5, 55.8 and 62.6 kg ha$^{-1}$ respectively for the 3, 4, 5, and 6 m tillage width machines, while the $CO_2$ emissions related to the use of the rotary tillers were 101.9, 63.2 and 72.7 kg ha$^{-1}$ of $CO_2$, for the machines with 3.2 m, 4.0 m and 5.2 m tillage width, respectively.

As to the operative performance of the combined cultivator and of the offset disk harrow, the test results showed a decrease of power and energy requirements proceeding from the conventional to the more conservative implement. Comparing the results of the 4.0 m width power harrow with the disk harrow, with the same working width and depth, the latter seems to represent a valid alternative to the first, allowing a considerable reduction in energy request per soil surface unit (42%), per unit of tilled soil volume (38%), and in fuel consumption per surface unit (40%). With respect to the 4.0 m rotary tiller (Table 4), the reduction in said parameters is even more evident, equal to 62%, 44% and 55%, respectively. The $CO_2$ emissions due to the conservation tillage implements ranged from 28.6 kg ha$^{-1}$ for the disk harrow to 63.7 kg ha$^{-1}$ for the combined cultivator. The conversion of the conventional tillage (based on power harrows and rotary tillers) to the reduced tillage (disking or combined tillage) could lead to reductions in overall $CO_2$ emissions. A seedbed prepared, in two passes, with primary tillage operated by the tested combined cultivator

and by a subsequent disking secondary tillage, will release a $CO_2$ amount equal to only 92.3 kg ha$^{-1}$.

### 3.2. Statistics

PCA results—Table 5 shows the eigenvalues and their relative variance explained by the principal components (PC) identified for PH and RT.

**Table 5.** Results of the Principal Component Analysis (PCA) carried out separately for the power harrow and the three rotary tillers considering specific parameters.

| PC | PH | | RT | |
|---|---|---|---|---|
| | Eigenvalue | % Variance | Eigenvalue | % Variance |
| 1 | 6.04 | 46.50 | 10.071 | 77.47 |
| 2 | 4.87 | 37.46 | 2.929 | 22.53 |
| 3 | 2.09 | 16.05 | - | - |

Diagrams of Figure 2 show the relationship between the variables and the first two principal components and among different components.

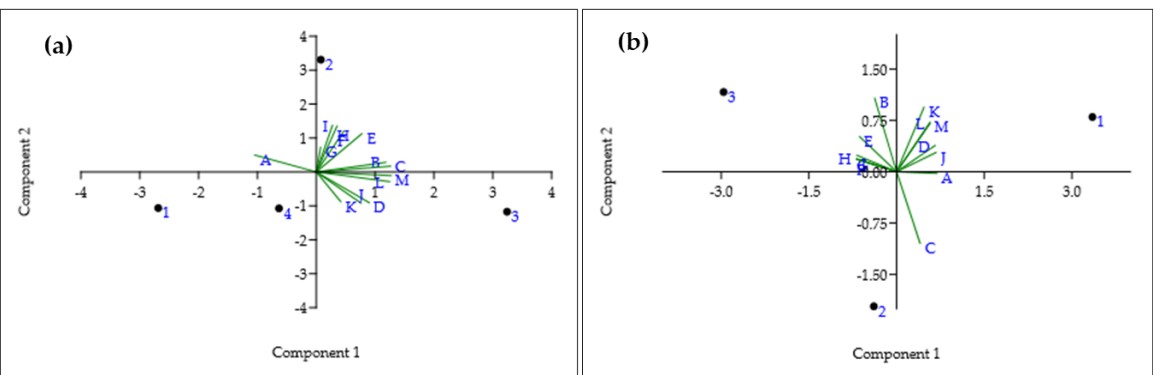

**Figure 2.** Diagrams of the relationship between the variables (letters A to L) and the principal components (PC1 and PC2) for the power harrows (**a**) and rotary tillers (**b**).

More detailed information is provided in Figure 3 by the correlation coefficients between each considered variable and the principal components of Table 5. Moreover, what seems more interesting is the presence or absence, of sign concordance among the coefficients of different variables, which indicates if they vary in concordance or in opposition.

Figure 3 suggests some relationship between soil characteristics and machines' performances, mainly in the PC1 in both PH and RT datasets. Considering the PH coefficients, in PC1 (46.5% of the total variability) there is sign concordance between all dynamic parameters and the soil humidity and dry bulk density, while the Cone Index is discordant. The higher correlations are observed for "Fuel consumption per hour per working width unit", "Specific traction force", "Energy losses for transmission" and "Total energy losses". The Cone Index was measured before the ploughing which preceded the secondary tillage. Moreover, it resulted to be quite constant in the tests with the four power harrows. Thus, the soil residual compaction in clods could become less effective in contrasting the rotation of the working tools than the humidity and the intrinsic density of the soil.

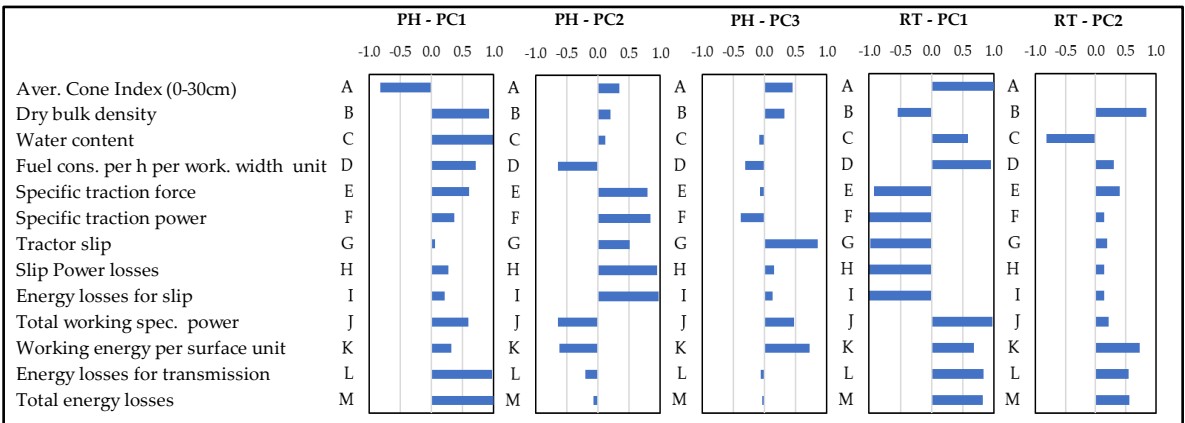

**Figure 3.** Correlation coefficient between each variable considered in the PCA and the principal components (PC) identified for the power harrow (PH) and the rotary tillers (RT, A to L).

These were measured before the harrowing and directly contributed to determining the magnitude of fuel consumption, working power and energy and of the energy losses. As to PC2 (37.5% of total variance), its correlation with soil parameters appears weaker, while it is stronger and concordant with (and among) the variables "E" to "I", inherent to the traction force and slip. The average values observed for such variables and their weight on the energy balance resulted however relatively low (Table 4). The correlations between PC3 and variables are less evident.

Observing the RT coefficients (Figure 3) in PC1 (77.5% of the total variance) we can note that, among soil variables, the "Cone Index" and the humidity are positively correlated to "Fuel consumption per hour per working width unit", "Total working specific. Power", "Working energy per surface unit", "Energy losses for transmission" and "Total energy losses", while the soil density is discordant in this case. Differently from the power harrows tests, the Cone Index then showed higher variability as well as the humidity (Table 4). This seemed to directly affect the trend of the aforementioned parameters, probably due to the direct way the clods are attacked by working tools, operated at lower rotor speed than the power harrows, but at higher torque values. It can be noted that the variables "E" to "I" have a negative sign, which means they decrease at increasing Cone Index and water content, as expected, because the pushing action of the L-shaped tools, exalted by soil compaction and, within certain limits, by the humidity, causes reduction of the traction force and of the slip (near 0).

Correlation results—The trend described by the PCA and the relative observations are confirmed by the Pearson test carried out on the performance parameters vs. soil parameters. The test was applied on couples of parameters. Each couple was formed by one of the performance parameters considered in the PCA and one soil parameter, calculating the coefficient "$r$" and the probability "$p$" of uncorrelation for each couple. The results are reported in Table 6 for the power harrows and the rotary tillers.

ANOVA results—Table 7 reports the two datasets at comparison based on the variables of Table 3. The values of the standard deviation and CV testify to the presence of high variability in all variables within each dataset. Despite this, the ANOVA was carried out with the aim of verifying any significance of the differences between the results provided by the power harrows and the rotary tillers.

**Table 6.** Correlation between the soil conditions (A, B, C) and the dynamic–energetic parameters (D to M) observed in PH and RT tests. Are reported the correlation coefficients, r, and the probability, p, of uncorrelation (r > 0.7: double underlined characters; r < −0.7: underlined characters; r > 0.95: bold, double-underlined characters; r < −0.95: bold, underlined characters).

|  | Variables |  | D | E | F | G | H | I | J | K | L | M |
|---|---|---|---|---|---|---|---|---|---|---|---|---|
| PH | A | r | −0.94 | −0.25 | −0.18 | 0.52 | 0.17 | 0.22 | −0.50 | −0.16 | −0.90 | −0.86 |
|  |  | p | 0.06 | 0.75 | 0.82 | 0.48 | 0.83 | 0.78 | 0.50 | 0.84 | 0.10 | 0.14 |
|  | B | r | 0.43 | 0.70 | 0.39 | 0.43 | 0.49 | 0.43 | 0.58 | 0.41 | 0.85 | 0.90 |
|  |  | p | 0.57 | 0.30 | 0.61 | 0.57 | 0.51 | 0.57 | 0.42 | 0.59 | 0.15 | 0.10 |
|  | C | r | 0.65 | 0.70 | 0.50 | 0.05 | 0.38 | 0.32 | 0.47 | 0.19 | 0.95 | **0.98** |
|  |  | p | 0.35 | 0.30 | 0.50 | 0.95 | 0.62 | 0.68 | 0.53 | 0.81 | 0.05 | 0.02 |
| RT | A | r | 0.95 | −0.91 | **−0.99** | **−0.99** | **−0.99** | **−0.99** | **0.97** | 0.67 | 0.82 | 0.82 |
|  |  | p | 0.20 | 0.27 | 0.08 | 0.11 | 0.08 | 0.08 | 0.15 | 0.54 | 0.38 | 0.39 |
|  | B | r | 0.17 | 0.54 | 0.27 | 0.31 | 0.27 | 0.26 | 0.09 | 0.64 | 0.44 | 0.45 |
|  |  | p | 0.89 | 0.64 | 0.83 | 0.80 | 0.82 | 0.83 | 0.94 | 0.56 | 0.71 | 0.70 |
|  | C | r | 0.31 | −0.87 | −0.69 | −0.72 | −0.69 | −0.69 | 0.39 | −0.21 | 0.03 | 0.02 |
|  |  | p | 0.80 | 0.33 | 0.51 | 0.49 | 0.51 | 0.52 | 0.75 | 0.87 | 0.98 | 0.99 |

**Table 7.** PH and RT datasets that underwent ANOVA. The variables considered are reported in Table 3.

| Variables | Unit | Power Harrows | | | | | | Rotary Tillers | | | | | |
|---|---|---|---|---|---|---|---|---|---|---|---|---|---|
|  |  | PH-3 | PH-4 | PH-5 | Aver. | St dev | CV(%) | RT-2.3 | RT-4 | RT-5.2 | Aver. | St dev | CV (%) |
| A | MPa | 1.30 | 1.31 | 1.20 | 1.27 | 0.06 | 4.79 | 1.66 | 1.14 | 0.76 | 1.19 | 0.45 | 38.02 |
| B | $kg\,m^{-3}$ | 1309 | 1390 | 1420 | 1373.00 | 57.42 | 4.18 | 1200 | 1100 | 1376 | 1225.33 | 139.73 | 11.40 |
| C | % | 11.0 | 12.8 | 14.3 | 12.70 | 1.65 | 13.01 | 15.1 | 17.2 | 10.7 | 14.33 | 3.32 | 23.14 |
| D | $kg\,h^{-1}m^{-1}$ | 6.07 | 5.66 | 6.96 | 6.23 | 0.67 | 10.68 | 11.66 | 8.95 | 8.57 | 9.72 | 1.69 | 17.33 |
| E | $kN\,dm^{-3}$ | 0.13 | 0.17 | 0.16 | 0.15 | 0.02 | 14.40 | 0.02 | 0.03 | 0.07 | 0.04 | 0.02 | 55.40 |
| F | $kW\,dm^{-3}$ | 1.88 | 2.76 | 2.22 | 2.29 | 0.45 | 19.52 | 0.39 | 0.82 | 1.31 | 0.84 | 0.46 | 55.12 |
| G | % | 2.30 | 7.30 | 3.60 | 4.40 | 2.59 | 58.96 | 0.92 | 1.22 | 1.62 | 1.25 | 0.35 | 28.02 |
| H | kW | 0.13 | 2.91 | 0.95 | 1.33 | 1.43 | 107.12 | 0.02 | 0.28 | 0.58 | 0.29 | 0.28 | 95.32 |
| I | $MJ\,ha^{-1}$ | 0.46 | 7.16 | 2.01 | 3.21 | 3.51 | 109.23 | 0.11 | 0.63 | 1.21 | 0.65 | 0.55 | 84.56 |
| J | $kW\,m^{-1}$ | 13.82 | 12.99 | 21.49 | 16.10 | 4.68 | 29.10 | 34.73 | 32.47 | 31.88 | 33.03 | 1.50 | 4.55 |
| K | $MJ\,ha^{-1}$ | 143.9 | 128.5 | 227.5 | 166.6 | 53.3 | 31.0 | 431.2 | 292.2 | 347.8 | 357.1 | 69.9 | 19.6 |
| L | $MJ\,ha^{-1}$ | 16.46 | 22.44 | 37.93 | 25.61 | 11.08 | 43.26 | 51.71 | 22.88 | 28.07 | 34.22 | 15.36 | 44.90 |
| M | $MJ\,ha^{-1}$ | 16.60 | 25.35 | 38.88 | 26.94 | 11.23 | 41.67 | 51.73 | 23.16 | 28.65 | 34.51 | 15.16 | 43.93 |

Table 8 reports the results of the ANOVA only for the four variables with $p < 0.1$. The differences within groups (PH and RT) were never significant. The differences between PH an RT resulted to be highly significant for "E" (specific traction force, $p = 0.017$), for "F" (specific traction power, $p = 0.040$) and "J" (total working specific power, $p = 0.036$). These results were expected due to both the close-to-zero values of traction force and power and to the high PTO power values observed in the rotary tillers, with respect to the power harrows. Therefore, the differences in "F" (working energy for surface unit) were also consistent, albeit with lower significance ($p = 0.063$). Such results substantially allow generalizing the above considerations on the comparison between the 4 m power harrow and rotary tiller. As to the remaining variables (including soil parameters), the p values ranged from 0.32 to 0.75 and the differences between PF and RT never resulted to be significant, probably because of the masking by the intrinsic, high heterogeneity of the machines and the working conditions which caused high general variability.

**Table 8.** Results of the ANOVA for the variables with significant difference between PH an RT: (1) $0.05< p \leq 0.1$: bold characters; (2) $0.025 < p \leq 0.05$: bold, underlined characters; (3) $< p \leq 0.025$: bold, double underlined characters.

| Variables | Difference | Dev. | D.o.F. | Var. | F | p (same) |
|---|---|---|---|---|---|---|
| **E** | Within groups | 0.0014 | 2 | 0.001 | 2.135 | 0.319 |
| | Between groups | 0.0190 | 1 | 0.0190 | **57.76** | **0.017** |
| | Error | 0.0007 | 2 | 0.0003 | | |
| | Total | 0.0210 | 5 | | | |
| **F** | Within groups | 0.558 | 2 | 0.279 | 2.094 | 0.323 |
| | Between groups | 3 | 1 | 3.146 | **23.61** | **0.040** |
| | Error | 0.266 | 2 | 0.133 | | |
| | Total | 4 | 5 | | | |
| **J** | Within groups | 158.94 | 2 | 79.47 | 0.489 | 0.672 |
| | Between groups | 429.99 | 1 | 429.99 | **26.45** | **0.036** |
| | Error | 32.52 | 2 | 16.26 | | |
| | Total | 478.41 | 5 | | | |
| **K** | Within groups | 7958.3 | 2 | 3979.2 | 1.061 | 0.485 |
| | Between groups | 54,401.5 | 1 | 54,401.5 | **14.51** | **0.063** |
| | Error | 7498.0 | 2 | 3749.0 | | |
| | Total | 69,857.8 | 5 | | | |

### 3.3. Quality of Work

Table 9 reports the values of the most significant parameters referring to the interaction between soil and machines: soil coverage index (SCI), crop residues/biomass burying degree (BBD), soil surface roughness index (SRI), roughness reduction degree (RRD), clod-breaking index (CBI), cloddiness reduction degree (CRD) and seedbed quality index (SQI).

**Table 9.** Average values of the soil tillage quality parameters for each tested implement.

| Parameters<br>Implements | SCI [1] (%) | BBD (%) | SRI | RRD (%) | CBI | CRD (%) | SQI |
|---|---|---|---|---|---|---|---|
| PH-3 | 0.7 | 82.1 | 1.79 | 80.1 | 0.84 | 36.5 | 1.29 |
| PH-4 | 0.6 | 84.6 | 1.69 | 81.0 | 0.82 | 36.3 | 1.25 |
| PH-5 | 0.6 | 84.0 | 2.41 | 63.7 | 0.82 | 56.3 | 0.87 |
| PH-6 | 0.5 | 85.7 | 1.54 | 81.5 | 0.83 | 57.9 | 0.92 |
| RT-2.3 | 0.6 | 81.0 | 2.17 | 72.1 | 0.84 | 53.8 | 1.10 |
| RT-4 | 0.7 | 80.8 | 2.32 | 70.8 | 0.83 | 56.7 | 1.16 |
| RT-5.2 | 0.8 | 78.6 | 1.27 | 82.5 | 0.85 | 57.1 | 0.94 |
| DH | 4.7 | 38.0 | 4.86 | 32.8 | 0.60 | 54.7 | 0.43 |
| CC [2] | 10.5 | 87.8 | 3.50 | - | 0.66 | - | 0.81 |

[1] evaluated after tillage; [2] working on undisturbed soil.

As to the parameters considered for evaluating the quality of work, the implements operated by the tractor's PTO left the soil free from crop residues, as testified by the high values, over 80%, of the biomass burying degree (BBD). The best soil surface roughness index (SRI) after the secondary tillage and the consequent reduction degree (RRD) were provided by the wider power harrow (6 m width) and rotary tiller (5.2 m width), due to the levelling action of the rear roller. Regarding the pulverization of the soil tilled layer for preparing an optimal seedbed, the same machines best performed in clod breaking, providing the highest CBI and the related CRD values as a consequence of the short tines-cutting interval (58.0 mm) of the 6.0 m width power harrow and the long tillage pitch (67.0 mm) of the 5.2 m width rotary tiller. Therefore, the respective seedbed quality index (SQI) resulted very close to the unity (1.0), considered as the optimal value.

As to the soil tillage quality parameters of the implements with passive action working tools, the best biomass residues burying degree (BBD) was provided by the combined

cultivator, thanks to the action of the two series of disks with the opposite inclination for soil reversing and mixing with crop residues. Such an implement also provided good CBI and SQI values.

These data become important when considering the number of secondary tillage interventions needed after the ploughing to prepare the seedbed. The availability of reliable data on the achievable energy savings and on the quality of work, related to medium/long period agronomic considerations on soil characteristics and crops physiological needs, will contribute to the choice of the most suitable tillage systems for seedbed preparation, orienting, at the same time, the manufacturers in the development of more efficient and effective machines and users in the choice of the most proper tillage system.

Within conventional cereal production systems, the seedbed preparation of silty-clay soil after the primary tillage usually consists of a deep ploughing. If started during the autumn to allow the atmospheric agents to improve the pulverization of the soil aggregates during the winter, it can be completed with one or more passes with disk or tine harrows, combined with packer rollers. If—due to adverse environmental conditions—the ploughing precedes the spring sowing a few days, a suitable seedbed can be quickly achieved by using machines with working tools powered by the tractor PTO (only one passage at low speed with a twin rotor, power harrow or rotary tiller) [56], or machines with passive action tools, tractor-drawn, such as disk harrows or cultivators (at least 2 passages at high speed). Currently, the most used rotary implements in conventional secondary tillage are rotary tillers and especially power harrows because they avoid the formation of a tillage pan and may be used at higher forward speeds. These machines are able to obtain a high percentage of soil aggregate size of 10 mm, which is a generally accepted target for a suitable seedbed.

The seedbed preparation in conservation tillage systems, on undisturbed soil, is generally limited to the superficial layer: the machines generally operate on crop residues and are equipped with working tools such as straight shanks, to disrupt compacted subsurface layers, and with disks and/or rollers capable of cutting the residues, pulverizing, and leveling the upper layer of soil. These machines, even combined together in a single implement and not powered by the tractor PTO but with passive action on the soil, are capable to prepare, in a single step, a good quality seedbed, for the sowing of an autumn-winter cereal, with consequent energy savings, and to avoid soil compaction [57,58].

The improvement of the operational efficiency of tractor–machine coupling deserves special consideration. It can be improved by maximizing the work output or reducing fuel consumption. Potential savings could be achieved with the gear-up throttle-down practice, which involves reducing engine speed and shifting to a faster gear to maintain the suitable working speed and implement efficiency [59]. The proper matching of an implement to a tractor is another method of increasing operational efficiency, especially for heavy drawbar work [60]. For a soil tillage machine, the required traction force varies as a function of the travel speed and of the operating tillage depth and width. The operating cost for any given implement could be minimized either by optimizing the working speed or the tillage width. Furthermore, the choice of an energy-efficient implement, such as a combined cultivator instead of a mouldboard plough or a subsoiler before secondary tillage, can reduce the tillage energy requirement [61].

## 4. Conclusions

The choice of the best type of intervention for the preparation of a seedbed depends on the physical characteristics of the soil, the type (ploughing or subsoiling) and depth of the primary tillage, the amount and the type of crop residues present on the surface and especially the type of crop to be planted which determines the necessary level of seedbed pulverization.

The field tests aimed at assessing working time, energy requests and quality of work, allowed to recommend the utilization of power harrows and rotary tillers, with working tools powered by the tractor PTO, for crops (such as maize, soybean, sugar beet) that require uniform seed spacing in the row, according to the cultivation standard requirements. The

high degree of soil pulverization and leveling produced by said machines ensures more stability to the seed drill during the seed placement in the furrow increasing the sowing precision.

Despite the general high variability caused by the different soil conditions and machine models, the test results evidenced relevant differences between the operative performances of the two typologies of tested machines. As a matter of fact, the power required at the tractor's PTO and the energy requirements per surface unit observer for the rotary tillers are significantly higher than for power harrows. However, the high energy demand by both machine typologies is compensated by the possibility of obtaining a suitable seedbed in a single pass, especially if characterized by great working width, to reduce the operative working time and the soil compaction due to repeated passages of the machines. On the other hand, the seedbed quality was rather different: the rotary tillers determined a high degree of soil pulverization which resulted to be homogeneous in the whole tilled layer. Using power harrows determined a lower degree of soil pulverization with larger diameter clods distributed in the superficial layer and consequent higher soil roughness after the harrowing.

On the other hand, the utilization of machines with static action tools appears more advantageous for the autumn–winter cereals that do not require high accuracy in seed placement (both for the large number of seed distributed per unit area, and for the capacity of bunching) and, therefore, do not require of high soil pulverization, also considering the superficial root system of these crops. In this case, a certain roughness can be maintained on the soil surface, especially in a rainy environment, even in the presence of reduced cloddiness, contributing to protecting the soil against the action of pelting rain, the resulting risks of superficial erosion and the formation of superficial crust, which would require further interventions. These conditions were well provided by the tested conservation implements (the disks harrow and the combined cultivator), which also required very low energy inputs. The first required a previous deep primary tillage and at least two successive steps to prepare a suitable seedbed for wheat planting, while the combined cultivator (which assemble straight shanks, disks and rear packer roller) allowed the deep working and a contemporary superficial fragmentation of the soil to prepare the seedbed in a single step, even in the presence of crop residues and/or biomass weed. A more pulverized seedbed, if needed, could be achieved by means of a following secondary tillage with a 5 m power harrow, capable to cover two cultivator passages, with high energy savings with respect to the traditional system based on the harrowing after ploughing.

**Author Contributions:** Conceptualization, R.F.; methodology, R.F. and D.P.; validation, R.F., and D.P.; formal analysis, R.F.; investigation, D.P.; resources, R.F.; data curation, R.F. and D.P.; writing—original draft preparation, R.F.; writing—review and editing, D.P. and P.S.; supervision, P.S. All authors have read and agreed to the published version of the manuscript.

**Funding:** This work was supported by the Italian Ministry of Agriculture (MiPAAF) under the AGROENER project (D.D. n. 26329, 1 April 2016)—http://agroener.crea.gov.it/.

**Institutional Review Board Statement:** Not applicable.

**Informed Consent Statement:** Not applicable.

**Data Availability Statement:** The data presented in this study are available on request from the corresponding author.

**Acknowledgments:** We thank Renato Grilli, Stefano, Benigni, Cesare Cervellini and Gino Brannetti for their precious technical support during the realization of the field tests.

**Conflicts of Interest:** The authors declare no conflict of interest.

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
