# Peer review of "Conventional and Conservation Seedbed Preparation Systems for Wheat Planting in Silty-Clay Soil"

_sustainability, doi:10.3390/su13116506_

Round 1

Reviewer 1 Report

The manuscript is relatively large in number of pages. However, the scope of the manuscript may be justified by the amount of quality information contained.

Table 3., Table 4. - Water content: weight percentage or volume percentage? It is appropriate to add.

Figure 2. - Diagrams are difficult to read. It is advisable to enlarge them.

Author Response

Comments and Suggestions for Authors

The manuscript is relatively large in number of pages. However, the scope of the manuscript may be justified by the amount of quality information contained.

Table 3., Table 4. - Water content: weight percentage or volume percentage? It is appropriate to add. 

Answer: The water content has been specified

Figure 2. - Diagrams are difficult to read. It is advisable to enlarge them.

Answer: The figure 2 has been enlarged

Reviewer 2 Report

The central idea of the work - Conventional and conservation seedbed preparation systems for wheat planting in silty-clay soil.

The authors of the study noticed that the conventional preparation of the seedbed was performed by deep and light plowing and the cultivation of the topsoil, which prepared the seedbed in one pass. They described soil maintenance methods with minimal or no tillage using passive implements requiring a minimum of two passes. The authors conducted field research consisting in assessing the energy and energy needs of conventional and conservation machines, as well as the quality parameters of soil cultivation in terms of the possibility of preparing an optimal substrate for wheat planting. The authors noted that the results of the research showed significant differences in fuel consumption, power and energy demand with rotary cultivators is much greater than with rotary harrows. Another conclusion is the observation of a decrease in the analyzed parameters, going from conventional to more conserving tillage machines, and that the better quality of the soil for sowing was ensured by rotary tillers. 

The obtained data is very interesting and brings new knowledge in the field of soil preparation for sowing.

In my oppinion scientific importance of the paper is significant.

The structure of the article is very transparent. Proposed methodology and conclusion are good discribe.

Note to the authors of the work:
In line 166 there is 106 CO and it should be 106oC. Items 38, 52 and 53 from the reference list were not noticed in the paper.

Author Response

REVIEWER 2

Open Review

(x) I would not like to sign my review report

( ) I would like to sign my review report

English language and style

( ) Extensive editing of English language and style required

( ) Moderate English changes required

( ) English language and style are fine/minor spell check required

(x) I don't feel qualified to judge about the English language and style

                Yes         Can be improved            Must be improved         Not applicable

Is the content succinctly described and contextualized with respect to previous and present theoretical background and empirical research (if applicable) on the topic?

                (x)          ( )           ( )           ( )

Are the research design, questions, hypotheses and methods clearly stated?

                (x)          ( )           ( )           ( )

Are the arguments and discussion of findings coherent, balanced and compelling?

                (x)          ( )           ( )           ( )

For empirical research, are the results clearly presented?

                (x)          ( )           ( )           ( )

Is the article adequately referenced?

                (x)          ( )           ( )           ( )

Are the conclusions thoroughly supported by the results presented in the article or referenced in secondary literature?

                ( )           ( )           ( )           ( )

Comments and Suggestions for Authors

The central idea of the work - Conventional and conservation seedbed preparation systems for wheat planting in silty-clay soil.

The authors of the study noticed that the conventional preparation of the seedbed was performed by deep and light plowing and the cultivation of the topsoil, which prepared the seedbed in one pass. They described soil maintenance methods with minimal or no tillage using passive implements requiring a minimum of two passes. The authors conducted field research consisting in assessing the energy and energy needs of conventional and conservation machines, as well as the quality parameters of soil cultivation in terms of the possibility of preparing an optimal substrate for wheat planting. The authors noted that the results of the research showed significant differences in fuel consumption, power and energy demand with rotary cultivators is much greater than with rotary harrows. Another conclusion is the observation of a decrease in the analyzed parameters, going from conventional to more conserving tillage machines, and that the better quality of the soil for sowing was ensured by rotary tillers.

The obtained data is very interesting and brings new knowledge in the field of soil preparation for sowing.

In my oppinion scientific importance of the paper is significant.

The structure of the article is very transparent. Proposed methodology and conclusion are good discribe.

Note to the authors of the work:

In line 166 there is 106 CO and it should be 106oC. Items 38, 52 and 53 from the reference list were not noticed in the paper.

Answer: in line 166 has been reported the right indication (106°C) and items 38, 52 and 53 has been noticed in the text.

Reviewer 3 Report

General Comment: The research is reasonably interesting, although it has to be included in the category of “case study”, more correctly. The paper is well structured and with a broad analysis but is excessively focused in introduction section. I suggest to add most studies that showed the correct application in international contest on sustainability and environmental aspects.

Abstract: Abstract need a partial revision to improve the comprehension of the work. I suggest to add the aim of the study. The authors described only methodology and excessively results aspect. Declare the aims and declare the most significant methods applied. I suggest to highlight the most important results and, briefly, present the main conclusions. CREA acronym is not correct to use in abstract section.

keywords: ok.

Introduction

Lines 34-35. “Seedbed preparation ……………………….. activities”. Please explain better this sentence and confirm this your sentence with references or

Line 45. “In Italy, the most common conventional method adopted…” How the authors can declare this data?

Lines 73-74. (minimum tillage ……. tillage) I suggest to eliminate the round bracket and rephrase the sentence with this detail.

Lines 78-80. “Due…..benefits”. I invite the authors to explain better this concept; it is not clear.

Line 93. Please Acronym CREA, used for the first time in the manuscript, should be explained.

Lines 93-106. “The Agricultural Machinery…………………. reports”. I suggest to move in M &M paragraph.

In several paragraphs the authors describe the activities as an academic lesson respect a research; some sentences are obvious and can be deleted to facilities the comprehension of the text.

M&M

Line 175. tiller;

Lines 143-154. The same operator drove the farm tractors? In the same day were conducted the test with two tractors? Years of experience of driver?........

Line 168. …image of the soil confined by the frame was taken. Which camera was used? How many megapixel was calibrated?

Line 171. The penetrologger is a versatile instrument for in situ measurement of the resistance to penetration of the soil and not for the measure of the force needed for the penetration. I suggest to rephrase the sentence.

Line 175. Who is ENAMA?

Results and Discussion are correctly described

Conclusion

The conclusions repeat the results adding considerations useful for discussion paragraph. More future developments and conclusions should be considered. I suggest to modify partly this section; the conclusions should be written comparing objects of the study and the results obtained but the authors reported a discussion section.

In reference, some authors are underlined with link web.

Author Response

Reviewer 3

Open Review

(x) I would not like to sign my review report

( ) I would like to sign my review report

English language and style

( ) Extensive editing of English language and style required

( ) Moderate English changes required

(x) English language and style are fine/minor spell check required

( ) I don't feel qualified to judge about the English language and style

                Yes         Can be improved            Must be improved         Not applicable

Is the content succinctly described and contextualized with respect to previous and present theoretical background and empirical research (if applicable) on the topic?

                ( )           (x)          ( )           ( )

Are the research design, questions, hypotheses and methods clearly stated?

                (x)          ( )           ( )           ( )

Are the arguments and discussion of findings coherent, balanced and compelling?

                (x)          ( )           ( )           ( )

For empirical research, are the results clearly presented?

                (x)          ( )           ( )           ( )

Is the article adequately referenced?

                ( )           (x)          ( )           ( )

Are the conclusions thoroughly supported by the results presented in the article or referenced in secondary literature?

                ( )           ( )           ( )           ( )

Comments and Suggestions for Authors

General Comment: The research is reasonably interesting, although it has to be included in the category of “case study”, more correctly. The paper is well structured and with a broad analysis but is excessively focused in introduction section. I suggest to add most studies that showed the correct application in international contest on sustainability and environmental aspects.

Answer: we added papers that showed the correct application of the conservation tillage methods on sustainability and environmental aspects. The references were upgraded.

Abstract: Abstract need a partial revision to improve the comprehension of the work. I suggest to add the aim of the study. The authors described only methodology and excessively results aspect. Declare the aims and declare the most significant methods applied. I suggest to highlight the most important results and, briefly, present the main conclusions. CREA acronym is not correct to use in abstract section.

Answer:  the aim of the study has been added and CREA acronym has been eliminated. We think that the most important results and the main conclusion has been properly highlighted.

keywords: ok.

Introduction

Lines 34-35. “Seedbed preparation ……………………….. activities”. Please explain better this sentence and confirm this your sentence with references or

Answer: this sentence has been explained better and confirmed by two references.

Line 45. “In Italy, the most common conventional method adopted…” How the authors can declare this data?

Answer: We declare this statement according to an informal poll on some contractors working in central Italy.

Lines 73-74. (minimum tillage ……. tillage) I suggest to eliminate the round bracket and rephrase the sentence with this detail.

Answer: the sentence has been rephrased and the brackets has been eliminated.

Lines 78-80. “Due…..benefits”. I invite the authors to explain better this concept; it is not clear.

Answer: the sentence has been rephrased

Line 93. Please Acronym CREA, used for the first time in the manuscript, should be explained.

Answer: CREA acronym has been explained.

Lines 93-106. “The Agricultural Machinery…………………. reports”. I suggest to move in M &M paragraph.

In several paragraphs the authors describe the activities as an academic lesson respect a research; some sentences are obvious and can be deleted to facilities the comprehension of the text.

Answer: the paragraph from line 93 to 106 has been moved in M&M.

M&M

Line 175. tiller;

Answer: this suggestion is incomprehensible (no tiller in line 175)

Lines 143-154. The same operator drove the farm tractors? In the same day were conducted the test with two tractors? Years of experience of driver?........

Answer: Only an operator drove the tractors. He has twenty years of experience in driving tractors during field tests. In the same day has been conducted the tests with only a tractor.

Line 168. …image of the soil confined by the frame was taken. Which camera was used? How many megapixel was calibrated?

Answer: in the line 185 we reported the used digital camera and the calibrated megapixel.

Line 171. The penetrologger is a versatile instrument for in situ measurement of the resistance to penetration of the soil and not for the measure of the force needed for the penetration. I suggest to rephrase the sentence.

Answer: the sentence has been rephrased

Line 175. Who is ENAMA?

Answer: ENAMA acronym has been explained.

Results and Discussion are correctly described

Conclusion

The conclusions repeat the results adding considerations useful for discussion paragraph. More future developments and conclusions should be considered. I suggest to modify partly this section; the conclusions should be written comparing objects of the study and the results obtained but the authors reported a discussion section.

Answer: In our opinion, the section Conclusion combines the obtained results with topics for discussion useful to evaluate future possible developments. At the end of the Conclusion section, we advise an intervention with a 2.5 m width combined implement followed by a secondary tillage by means of a 5  m power harrows, to obtain a suitable seedbed for wheat planting.

In reference, some authors are underlined with link web.

Answer: In reference, the underlining of some author was eliminated
